# Analysis of Hematological Traits in Polled Yak by Genome-Wide Association Studies Using Individual SNPs and Haplotypes

**DOI:** 10.3390/genes10060463

**Published:** 2019-06-17

**Authors:** Xiaoming Ma, Congjun Jia, Donghai Fu, Min Chu, Xuezhi Ding, Xiaoyun Wu, Xian Guo, Jie Pei, Pengjia Bao, Chunnian Liang, Ping Yan

**Affiliations:** 1Animal Science Department, Lanzhou Institute of Husbandry and Pharmaceutical Sciences, Chinese Academy of Agricultural Sciences, Lanzhou 730050, China; 82101171175@caas.cn (X.M.); dkjcj@nwafu.edu.cn (C.J.); 82101175147@caas.cn (D.F.); chumin@caas.cn (M.C.); dingxuezhi@caas.cn (X.D.); wuxiaoyun@caas.cn (X.W.); guoxian@caas.cn (X.G.); peijie@caas.cn (J.P.); baopengjia@caas.cn (P.B.); 2Key Laboratory for Yak Genetics, Breeding, and Reproduction Engineering of Gansu Province, Lanzhou 730050, China

**Keywords:** single-marker GWAS, haplotype analysis, hematological traits, polled yak

## Abstract

Yak (*Bos grunniens*) is an important domestic animal living in high-altitude plateaus. Due to inadequate disease prevention, each year, the yak industry suffers significant economic losses. The identification of causal genes that affect blood- and immunity-related cells could provide preliminary reference guidelines for the prevention of diseases in the population of yaks. The genome-wide association studies (GWASs) utilizing a single-marker or haplotype method were employed to analyze 15 hematological traits in the genome of 315 unrelated yaks. Single-marker GWASs identified a total of 43 significant SNPs, including 35 suggestive and eight genome-wide significant SNPs, associated with nine traits. Haplotype analysis detected nine significant haplotype blocks, including two genome-wide and seven suggestive blocks, associated with seven traits. The study provides data on the genetic variability of hematological traits in the yak. Five essential genes (*GPLD1*, *EDNRA,*
*APOB*, *HIST1H1E*, and *HIST1H2BI*) were identified, which affect the HCT, HGB, RBC, PDW, PLT, and RDWSD traits and can serve as candidate genes for regulating hematological traits. The results provide a valuable reference to be used in the analysis of blood properties and immune diseases in the yak.

## 1. Introduction

Yak (*Bos grunniens*) is an iconic symbol of the Qinghai–Tibet Plateau and adjacent high-altitude areas [1]. This region is well known for its high-altitude, pristine natural environment, and extreme seasonal changes [2]. During the early Holocene period, the ancient Qiang people began to domesticate wild yaks [3,4]. While only a few animal species can survive in the Qinghai–Tibet Plateau, yaks thrive in the extreme conditions present in this area. They provide food such as meat and milk, a mode of transportation, shelter, and fuel for the residents of this high-altitude region [5,6]. As typical representatives of indigenous animals, yaks are characterized by a high degree of adaptability and low sensitivity to harsh ecological pressures [7,8]. However, due to relatively primitive breeding technology used by the herdsmen, and the difficulty or even lack of implementing immunizations and other measures preventing epidemics, yaks are very susceptible to parasites, which has caused significant economic losses to the yak-breeding industry.

Blood cells have a critical function in immune response and resistance to diseases [9,10]. They include three major cell types: leukocytes (white blood cells, WBCs), erythrocytes (red blood cells, RBCs), and platelets. WBCs are involved in the phagocytosis of bacteria and disease prevention. RBCs are essential for oxygen transport within the organism and the removal of the byproducts of respiration. The platelets play an important role in blood coagulation, wound healing, inflammatory response, and organ transplant rejection [11]. Blood homeostasis is the property responsible for the maintenance of routine indicators, like the number, volume, and biological activity of peripheral blood cells, within a relatively narrow physiological range. In pathologic conditions, these indicators are frequently outside of the normal physiological range [12]. In humans, these traits are to a large extent inherited but also influenced by environmental factors, such as age, gender, obesity, smoking, and alcohol consumption. Genetic and environmental factors can produce significant differences among individuals [13,14,15,16]. Therefore, the hematological indicators are often used as clinical indicators reflecting the overall health of patients.

Genome-wide association studies (GWASs) represent a powerful tool for mapping genetic loci associated with common disease and quantitative traits. Hematologic parameters are important indicators reflecting the immune status and health of the human or animal organism and are used as biomarkers in GWASs [17]. In recent years, GWASs have been used in human studies to locate more than 60 quantitative trait loci (QTLs) for blood traits among people of European ancestry, the Japanese and African Americans [18,19,20,21,22,23,24,25]. In the field of animal husbandry, pigs have been bred for disease resistance earlier than other species. Many QTLs affecting blood parameters traits in this species were also identified. A search of the animal QTL database [26] yielded 2482 QTLs affecting pig blood traits. Using whole genome-wide scanning in the White Duroc*Erhualian F2 resource population, Yang and coworkers located QTLs that affect the characteristics of WBCs and platelets on the *Sus scrofa* chromosomes (SSCs) 2, 7, 8, 12, 13, 15, and X [27]. Zou and collaborators identified QTLs affecting RBC traits at three growth stages in SSCs 1, 2, 3, 4, 5, 6, 7, 8, 9, 10, and 13 [28]. Wang and collaborators used the porcine SNP60 gene chip for GWASs of 18 hematological traits (seven leukocyte traits, seven red blood cell traits, and four platelet traits) affecting piglet vaccine immunization. After the alignment based on multiple tests to adjust for each property characteristic of leukocytes, erythrocytes, and platelets, chromosomes 10, 24 and 77 on the SNP were found to be significant [29]. To the best of our knowledge, the GWAS approach was not used for bovine blood traits. The only available information is related to validation tests on haplotypes of related *Leptin* genes [30].

The current study employs the bovine HD SNP chip to locate the QTLs of blood traits in Polled yaks by GWAS analysis. The objective of this work is to provide crucial preliminary information necessary for the identification of genes affecting blood traits and causal mutations in yak.

## 2. Materials and Methods

All yaks involved in this study were in compliance with the guidelines for the care and use of laboratory animals issued by the State Council of the People’s Republic of China. Additionally, the current investigation was approved by the Animal Management and Ethics Committee of the Lanzhou Institute of Animal Science and Veterinary Medicine, Chinese Academy of Agricultural Sciences (Permit No. SYXK-2016-0039)

### 2.1. Animal Blood Sample Collection

The cohort of yaks used in this study was from the Ashdan Mountain area of Qinghai Province, China. Blood samples, 5 mL each, were collected from the jugular vein of 315 yaks. The animals had no genetic relationship to avoid inclusion of yaks with the same pedigree. DNA was extracted by the phenol–chloroform method [31], and the isolated DNA was dissolved in a Tris-EDTA (TE) buffer solution at pH = 8.0. The concentration and quality of DNA were measured by the NANODROP 1000 Nucleic Acid Protein Analyzer (Thermo Fisher Scientific Inc., Waltham, MA USA). The ratio of A 260/280 between 1.8 and 2.0 was considered to reflect acceptable quality. Qualified DNA samples were diluted to yield the concentration of 50 ng/μL and stored at −20 °C for subsequent SNP chip typing.

Standard sets of hematological data were recorded using the Sysmex pocH-100iV Diff whole blood analyzer (Sysmex Corporation, Japan) within 24 h of specimen collection at the Lanzhou Institute of Husbandry and Pharmaceutical Sciences of Chinese Academy of Agricultural Sciences. The remaining portion of the blood sample was stored at −20 °C until DNA extraction (see above). To perform single-marker GWASs, 15 blood traits were used. They included 8 erythroid traits: hematocrit (HCT), hemoglobin (HGB), mean corpuscular hemoglobin (MCH), mean corpuscular hemoglobin concentration (MCHC), mean corpuscular volume (MCV), red blood cell count (RBC), red blood cell volume distribution width-SD (RDW-SD), and red blood cell volume distribution width (RDW). Three leukocyte traits were also analyzed: lymphocyte count (LYM), white blood cell count (WBC), and medium white blood cell count (OTHR). The remaining four traits were related to the platelets and included platelet distribution width (PDW), platelet count (PLT), platelet-large cell ratio (P-LCR), and mean platelet volume (MPV). The presence of correlations between the 15 hematological parameters analyzed was established using the R psych package (http://personality-project.org/r/psych.manual.pdf).

### 2.2. SNP Array Genotyping and Quality Control

Genomic DNA was processed according to Illumina’s chip protocol and the chip was scanned using the iScan instrument (Illumina Inc., San Diego, CA, USA). The results of the chip scan were analyzed with GenomeStudio (version 2011.1) software. Quality control was performed using PLINK (version 1.07) [32] and results with a parameter of detection rates >95.0% and minor allele frequencies >0.01, HWE (*p*-value < 1×10^−5^), were excluded. Moreover, yaks with more than 10% of missing genotypes or with Mendelian errors greater than >5% were excluded. Only SNP markers that were consistent with family separation rules have been included in subsequent statistical analyses.

### 2.3. Statistical Analyses

The Bonferroni calibrated multiple test [33] was used to determine the significance threshold, with a genomic significant level threshold of 0.05 per effective SNP locus and a chromosomal significant horizontal threshold of 1 per effective SNP locus. For the purpose of this study, the corresponding thresholds were set as 9.5 × 10^−6^ (0.05/104,679) and 4.7 × 10^−7^ (1/104,679).

### 2.4. Single-Marker GWAS

For each SNP, the linear trend of alleles and phenotypic traits was assessed by the general linear mixed model [34,35]. The linear model contains polygenic random effects, and a variance–covariance matrix to form a structural Q matrix. The Q matrix was computed using admixture [36]. Since the CV value (Corss validation) was found to be the smallest when K = 3, therefore K = 3 was chosen for the Q matrix.

The model was described as follows:*y* = *Xb*_i_ + *Q*v + *e*
where *y* is the phenotype, *b*_i_ is the regression coefficient of phenotype on SNP genotypes, *X* represents the vector of the SNP genotype indicators and assumes values of 0, 1, or 2 corresponding, respectively, to the three genotypes 11, 12 and 22, v is a variable dependent on population structure, Q is the corresponding principal components matrix, and e is the vector of residual errors with *e*, *N* (0, Iσ e2). The single-marker GWASs were conducted using the Tassel 5.2.43 software package [35].

### 2.5. Haplotype Analysis

Individual haplotype blocks and haplotype frequency were detected using the standard expectation maximization (EM) [37] algorithm. Reference thresholds and partitioning criteria for SNPs within closely linked haplotypes were established following the recommendations of the user manual of the PLINK software [32]. The recessive multi-alleles were used as a random effect in the association study model. The haplotype analysis model used in the current research is similar to that employed in the single-marker model, except for the replacement of the SNP effect in the single-marker GWAS by the haplotype effect.

## 3. Results

### 3.1. SNP Characteristics and Phenotype Statistics

The mean, standard deviation and coefficient of variation (CV) for 15 phenotypic observations of blood traits in the experimental population are presented in Table 1. The CV values ranged from a minimum of 2.92 to a maximum of 57.71 for MCHC and P-LCR, respectively.

After quality control, none of the animals had a genotyping call rate lower than 95%; all 315 yaks were included in the association analyses. A total of 42,276 SNPs failed the missingness test (GENO > 0.05), 653,185 SNPs had a minor allele frequency (MAF) of less than 0.05, and 8710 SNPs were severely departing from the Hardy Weinberg Equilibrium (HWE) (*p*-value less than 10^−5^). These SNPs were excluded from further analysis so that a total of 104,864 SNPs remained. Additional 185 SNPs, including unmapped SNPs, were also removed, yielding a total of 104,679 SNPs used for subsequent GWASs.

### 3.2. Loci for Erythrocyte Traits

Sixteen significant SNPs were found by using the single-marker GWAS method. They included nine suggested and seven genome-wide significant SNPs. The SNPs were identified for three erythrocyte traits, of which six were for HCT, five for HGB, and five for RBC (Table 2). Together, 16 single markers were detected which were located within five annotated genes. They include nine markers in the region of genes region, and one marker was located in a region 69,046 bp away from the nearest annotated gene. All identified markers were distributed on BTA (*Bos taurus* chromosome) 5, 16, 14, and 23. The most significant association of SNP was with the HCT trait (*p* = 6.21 × 10^−8^) (Figure 1).

In haplotype analysis, a total of six significant haplotype blocks, including two genome-wide and four suggestive blocks, were identified for four erythrocyte traits. One block each was associated with HCT, HGB, and RDWSD, while three blocks were associated with RBCs (Table 3). These significant SNPs were located on chromosome 23 (Figure 2). The most significant haplotype block was located in the *GPLD1* gene and was associated with the HGB trait.

### 3.3. White Blood Cell Counts

By single-marker GWAS, seven single suggestive SNPs were found for two white blood cell counts traits. Three of them were identified for LYM and four for OTHR. The most significant (*p* = 7.71 × 10^−7^) marker, bovine HD0100040184 was associated with OTHR and was located in the ATP2C1 gene on chromosome 1 (Figure 1).

Haplotype analysis revealed only 1 significant haplotype block, which was associated with OTHR traits. The most significant annotation blocks located in the LOC785941 gene on chromosome 8 was also associated with OTHR traits (Figure 2).

### 3.4. Platelet Traits

Single-marker GWAS detected 14 significant suggestive SNPs. All of them were associated with PLT. Expected PLT traits markers were included within genes or within the nearest regions of genes; other trait makers were outside the gene region. The most significant (*p* = 3.87 × 10^−7^) PLT trait marker was bovine HD2100007261, located in the SH3GL3 gene on chromosome 4 (Figure 1).

Haplotype analysis detected two haplotype blocks for PLT. Both blocks were located on chromosome 23 at position from 31,565,453 to 31,686,075 bp position (Figure 2).

## 4. Discussion

The present investigation employed a single-marker and haplotype GWAS method based on the BovineHD BeadChip to analyze 15 hematological traits in polled yaks. The accumulated data resulted in the identification of 43 significant SNPs by single-marker GWAS, including 35 suggestive and eight genome-wide significant SNPs, as well as nine significant haplotype blocks, including seven suggestive and two genome-wide blocks, by haplotype analysis. The combination of single-marker and haplotype analyses revealed significant associations of 15 hematological quantitative traits in comparable genomic regions. However, the distribution of genomic *P* values obtained with these two methods were slightly different. This apparent lack of consistency points to the fact that the efficacy of the analysis is data dependent. Additionally, the results obtained with the single-marker method were characterized by more significant *p*-values than those acquired with the haplotype method. The reason for this discrepancy may be due to the close relationship between continuous SNPs located in the same region; clustering of SNPs may reduce the number of significant correlations. It is often difficult to weigh between increasing LD and increasing freedom which results in reduced detection efficiency. Furthermore, the LDs distributed in the genome are not uniform, i.e., in some places, the LD is high, whereas in others places the LD is low, so we compared the two methods to capture more associated SNPs.

The current investigation represents the first GWAS of hematological traits in the yak. Previous studies using this methodology in livestock were focused on hematological traits in pigs. By employing a similar method, Zhang and coworkers have determined that four SNPs on SSCs 2 have pleiotropic effects when single-marker analysis was used and are located at 24,777,963 bp on chromosome 7 when haplotype analysis was employed [38]. Seven genes have been selected as potential candidates after SNP annotations and expression variation were analyzed. Zhang studied selective White Duroc 6 Erhualian F2 intercross in three growth stages and found 185 significant SNPs for 18 hematological traits. In the GWAS analysis of the dairy cow, hematologic traits were not addressed, but the association of leptin with immunological [39] and hematological traits was identified. The study of Orrù and colleagues [30] revealed that haplotype polymorphism affects the hematologic variables of the dairy cow during the perinatal period. Our study identified several novel SNPs affecting blood traits. However, they did not overlap with significant loci of the same traits in other species. The possible reasons for this discrepancy include differences among species, the use of cattle chips that yielded only 10,486 SNPs after the quality control caused insufficient marker density, and the genetic complexity of blood cells.

The results of this GWAS showed a significant association pattern of erythrocyte phenotypes. The Manhattan plot pattern was similar for HGB, HCT, and RBC and the three erythrocyte traits which shared common regions at BTA23, from 32,982 Mb to 32,983 Mb, containing two SNPs (BovineHD2300009627 and BovineHD2300009628). Both of these parameters are directly related to Erythrocytes, respectively correlation analysis of hematological traits by comparison (Appendix A). Moreover, the results indicated a high correlation between the three erythrocyte traits (r1 = 0.96, r2 = 0.93, r3 = 0.94; *p* < 0.001). A significant degree of correlation implies that the QTL on BTA23 might be pleiotropic. However, this association was not found in leukocyte and platelet traits.

By applying a single-marker analysis, the current study identified 43 significant SNPs on 12 different chromosomes associated with 15 hematological traits (Appendix A). Together, 15 genes occupying regions from 63 to 77,309 bp were annotated. The subsequent verification of the function of these genes resulted in the selection of three genes as potential candidates. They included endothelin receptor type A (EDNRA), apolipoprotein B (APOB), and glycosylphosphatidylinositol-specific phospholipase D1 (GPLD1)—all functionally associated with blood-related cells or immune function.

The SNP BovineHD1700003099 located in the gene explained 13.35% of phenotypic variants of PDW (Table 2). EDNRA, a homologous isomer of the endothelin receptor, belongs to the rhodopsin family of the G protein-coupled receptor superfamily [40]. EDNRA is an ET-1 selective receptor, expressed mostly in vascular smooth muscle cells. Activation of EDNRA results in prolonged vasoconstriction. The endothelin receptors are closely related to the onset and development of cardiovascular and kidney diseases, diabetes mellitus, autoimmune diseases, and cancer [41,42,43]. The *EDNRA* gene is also involved in the vasoconstriction mechanism, which is closely related to pulmonary hypertension [44]. In addition, the results indicated that 14 markers were significantly correlated with the PLT traits, and the association of the BovineHD 1100022374 marker was most significant (*p* =1.68 × 10^−6^). Moreover, all these markers were located within the *APOB* gene. The expression of the *APOB* gene is positively correlated with the degree of atherosclerosis, and the composition and function of platelets in patients with hyperlipidemia is altered. These changes may be related to the effect of circulating lipoproteins on platelet phenotype [45]. Since yaks are chronically exposed to low atmospheric pressure and hypoxia, the compensatory increase in the number of erythrocytes leads to a higher blood viscosity, which results in the disorder of lipid metabolism and an elevation in blood lipid levels. In this regard, hypoxia promotes the formation of atherosclerotic plaque through the increase of permeability of vascular walls for lipids [46]. There is a coordination relationship between the platelet activation mechanism and the cholesterol-rich microdomain [47]. In the species of cattle with close relatives of yak, the base mutation and insertion mutation of the APOB gene can be found to cause cholesterol deficiency in the individual [48,49]. Furthermore, the data obtained in the present study demonstrate that two significant markers (BovineHD4100016160, BovineHD2300009636) overlapped and fell into the *GPLD1* gene, which is also associated with erythrocyte traits. The product of the *GPLD1* gene, also known as GPI-PLD and PLD, belongs to the glycosyl phosphatidyl inositol (GPI) family of proteolytic enzymes. The concentration of GPLD1 in human serum can be as high as 5–10 μg/mL [50]. GPI has a variety of anchoring proteins widely expressed in the cardiovascular system. During the onset and development of atherosclerosis, GPI acts not only as a physiological receptor, but also activates signaling pathways leading to growth or apoptosis of vascular cells. The deficiency of the homologous complement increases the sensitivity of erythrocytes, becoming the most important factor in the pathogenesis of paroxysmal nocturnal hemoglobinuria. It has been documented that the overexpression of GPLD1 increases its ability to anchor CD55 and CD59 by nearly 50%, resulting in the inhibition of cell proliferation, and increase in the rate of apoptosis and complement-mediated cell death. These effects are attributed to the blockade of the CD55 and CD59 signaling pathway and the decrease or inhibition of the resistance to complement activity, constituting an important factor in the immune escape of cancer cells [51]. Together, these findings point to the involvement of the *GPLD1* gene in the pathways affecting the properties of red blood cells, making *GPLD1* a candidate gene controlling the characteristics of red blood cells.

In the present work, nine significant haplotype blocks located mostly in BTA8, BTA23, and BTA24 were identified by haplotype analysis. Among them, three significant blocks for RBC were found within two annotated genes, *GPLD1* and *TINAg*, located on BTA23. Also, RBC, HCT, and HGB were shown to have common significant blocks located on the *GPLD1* gene. Another erythrocyte trait, RDW-SD, was associated with the block at BTA23, from 3150 to 3156 Mb containing seven SNPs (BovineHD2300009102, BovineHD2300009103, BovineHD2300009105, BovineHD2300009110, BovineHD2300009112, BovineHD2300009113 and BovineHD2300009117), a region that also includes the *HIST1H1E* and *HIST1H2BI* genes. These genes are associated with the function of blood and immune cells. Kordbacheh and coworkers have demonstrated that histones promote erythrocyte aggregation and sedimentation in a concentration-dependent manner, increasing their osmotic fragility and lysis [52]. In addition, histones negatively interfere with the denaturation of red blood cells by reducing their ability to pass through an artificial spleen. In vivo, histones induce anemia, increase hemoglobin content in the spleen, and decrease the number of platelets and white blood cells within minutes.

## 5. Conclusions

In summary, our research provides important information about the genetic determinants of hematological traits in yaks and points out five genes, namely, *GPLD1*, *EDNRA*, *APOB*, *HIST1H1E*, and *HIST1H2BI*, affecting the HCT, HGB, RBC, PDW, PLT, and RDWSD traits, and serving as candidate genes for the future dissection of molecular mechanisms regulating hematological traits. The present investigation provides a valuable reference for the analysis of blood properties in the yak and immune diseases impacting this species.

## Figures and Tables

**Figure 1 genes-10-00463-f001:**
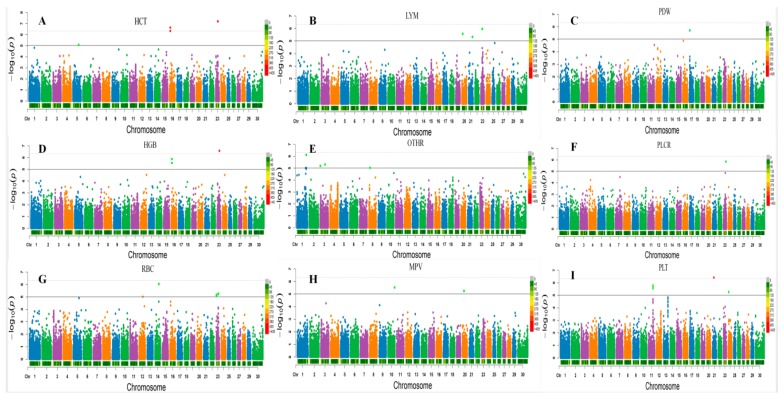
A portion of the GWAS results of hematological traits at a significant level in Polled yak. X axis indicates chromosome, Y axis indicates −lg (*p*-value), the solid line indicates genome-wide significant level and the dashed line indicates the chromosome-wide significance level, the bars below each chromosome are the density of the markers. RBC: red blood cell count. HCT: hematocrit; HGB: hemoglobin; OTHR: medium white blood cell count; PLT: platelet count; MPV: mean platelet volume; PLCR: platelet count; LYM: lymphocyte count; PDW: platelet distribution width.(**A**–**I**) Manhattan plot representing HCT, LYM, PDW, HGB, OTHR, PLCR, RBC, MPV and PLT traits, respectively.

**Figure 2 genes-10-00463-f002:**
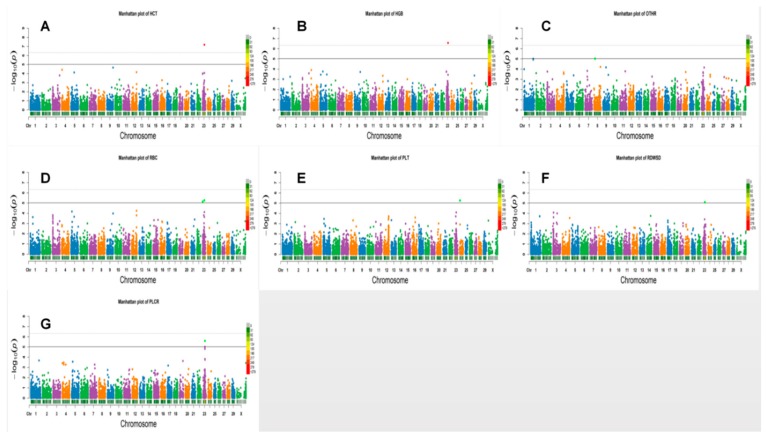
Manhattan plots for the haplotype analysis of hematological traits surpass the genome-wide significant threshold. Log 10 (1/*p*-value) values are shown for all SNPs that passed quality control. The numbers indicate the chromosomes in the genome. The solid line and dotted line denote the Bonferroni-corrected genome-wide and suggestive significant threshold, respectively. SNPs surpassing the genome-wide threshold are highlighted in red and SNPs reaching the suggestive threshold in green. The bars below each chromosome are the density of the markers. E.g., Manhattan of HCT means Manhattan map obtained by haplotype analysis of HCT traits. HCT: hematocrit; RBC: red blood cell count; HGB: hemoglobin; OTHR: medium white blood cell count; PLT: platelet count; RDW-SD: red blood cell volume distribution width-SD; PLCR: platelet count.(**A**–**G**) Manhattan plot representing HCT, HGB, OTHR, RBC, PLT, RDWSD and PLCR traits, respectively.

**Table 1 genes-10-00463-t001:** Descriptive statistics of 15 hematological traits in the yak population.

Traits	Abbreviation	Value 1 (No. ^2^)	CV ^3^
Mean corpuscular hemoglobin concentration (g/L)	MCHC	318.061 ± 9.281 (313)	2.92
Red blood cell volume distribution width-SD (fL)	RDWSD	31.427 ± 2.362 (315)	7.52
Mean corpuscular volume (fl)	MCV	42.725 ± 3.25 (315)	7.61
Red cell distribution coefficient of variation	RDWCV	0.187 ± 0.015 (315)	7.75
Mean platelet volume (fl)	MPV	7.146 ± 0.627 (171)	8.77
Mean corpuscular hemoglobin (pg)	MCH	13.611 ± 1.312 (315)	9.64
Red blood cell count (10^12^/L)	RBC	10.031 ± 1.266 (314)	12.62
Hematocrit (%)	HCT	0.429 ± 0.056 (309)	12.83
Hemoglobin (g/L)	HGB	136.217 ± 18.577 (314)	13.64
Platelet distribution width (%)	PDW	8.777 ± 1.314 (169)	14.97
White blood cell count (10^9^/L)	WBC	9.04 ± 2.197 (278)	24.31
Medium white blood cell count (10^9^/L)	OTHR	4.603 ± 1.505 (270)	32.69
Lymphocyte count (10^9^/L)	LYM	4.538 ± 1.588 (276)	34.99
Platelet count (10^9^/L)	PLT	326.193 ± 170.149 (311)	52.17
Platelet-large cell ratio (%)	PLCR	0.068 ± 0.039 (111)	57.71

^1^ Values are shown in mean ± standard deviation. ^2^ The numbers of recorded individuals are given in parentheses. ^3^ CV: coefficient of variation.

**Table 2 genes-10-00463-t002:** Associated SNPs and nearby candidate genes for hematological traits.

Traits ^1^	SNP ^2^	CHR ^3^	Position ^4^	*p* Value	VAR (%) ^5^	Nearest Gene ^6^	
						Name	Distance (bp)
**HCT ***	BOVINEHD0500023978	5	84784989	8.51× 10^−6^	7.44	IFLTD1	69,046
**HCT ****	BOVINEHD4100016160	23	32994655	6.21× 10^−8^	10.48	GPLD1	WITHIN
**HCT ****	BOVINEHD2300009636	23	32996521	6.55 × 10^−8^	10.38	GPLD1	WITHIN
**HGB ****	BOVINEHD4100016160	23	32994655	2.63× 10^−7^	9.48	GPLD1	WITHIN
**HGB ****	BOVINEHD2300009636	23	32996521	2.76 × 10^−7^	9.38	GPLD1	WITHIN
**RBC ***	BOVINEHD1400016753	14	60395118	9.25 × 10^−7^	7.52	LOC782927	WITHIN
**RBC ***	BOVINEHD2300009636	23	32996521	5.40 × 10^−6^	7.59	GPLD1	WITHIN
**RBC ***	BOVINEHD4100016160	23	32994655	5.84 × 10^−6^	7.59	GPLD1	WITHIN
**RBC ***	BOVINEHD2300001545	23	6375280	6.80 × 10^−6^	7.45	C23H6ORF142	42
**RBC ***	BOVINEHD2300001550	23	6383172	7.65 × 10^−6^	7.38	C23H6ORF142	42
**MPV ***	BOVINEHD1000028551	10	98690434	3.00 × 10^−6^	14.31	LOC788481	13,887
**MPV ***	BOVINEHD2000007009	20	23305743	5.68 × 10^−6^	11.73	IL31RA	63
**PDW ***	BOVINEHD1700003099	17	10778156	1.98 × 10^−6^	13.35	EDNRA	WITHIN
**PLT ***	BOVINEHD1100022362	11	77897626	1.68 × 10^−6^	7.29	APOB	WITHIN
**PLT ***	BOVINEHD2100007261	21	25044956	3.87 × 10^−7^	9.3	SH3GL3	11,156
**PLT ***	BOVINEHD2400002733	24	9663513	5.51 × 10^−6^	7.69	LOC538958	77,309
**LYM ***	BOVINEHD2000004120	20	12840434	2.75 × 10^−6^	9.13	LOC529061	WITHIN
**LYM ***	BOVINEHD2100017671	21	61024785	4.74 × 10^−6^	8.76	LOC781495	WITHIN
**LYM ***	BOVINEHD2300005782	23	21870423	1.11 × 10^−6^	8.48	MUT	68,056
**OTHR ***	BOVINEHD0100040184	1	140543231	7.71 × 10^−7^	10.17	ATP2C1	20,601
**OTHR ***	BOVINEHD0300001040	3	3461567	6.51 × 10^−6^	8.7	LOC100295170	2804
**OTHR ***	BOVINEHD0300021395	3	73543262	4.99 × 10^−6^	8.89	NEGR1	WITHIN
**OTHR ***	BOVINEHD0800005780	8	18417107	9.50 × 10^−6^	8.44	TUSC1	151,827

^1^ The abbreviations of hematological traits are given in Table 1, e.g., MCV is mean corpuscular volume. ^2^ The number of significant SNPs for each hematological trait. ^3,4^ Chromosomal locations and positions of the most significant SNP associated with hematological traits in Bos taurus UMD_3.1 assembly. ^5^ Interpretation rate of phenotypic variation of the marker. ^6^ The nearest annotated gene to the significant SNP. The annotated gene database is from http://asia.ensembl.org/index.html. SNP designated as in a gene or distance (bp) from a gene region in Bos taurus UMD_3.1 assembly. ** Genome-wide significant. * Chromosome-wide significant.

**Table 3 genes-10-00463-t003:** Associated haplotype blocks and nearby candidate genes for hematological traits.

Traits ^1^	NSNP ^2^	CHR^3^	START ^4^	END ^5^	SNP1 ^6^	SNP2 ^7^	VAR (%) ^8^	*p* Value	Nearest Gene ^9^	
									Name	Distance (bp)
**RBC ***	2	23	6045141	6051410	BovineHD2300001442	BovineHD2300001444	7.41	7.16 × 10^−6^	TINAG	WITHIN
**RBC ***	2	23	6045141	6051410	BovineHD2300001442	BovineHD2300001444	7.39	7.45 × 10^−6^	TINAG	WITHIN
**RBC ***	2	23	32982199	32983832	BovineHD2300009627	BovineHD2300009628	7.58	5.40 × 10^−6^	GPLD1	WITHIN
**HGB ****	2	23	32982199	32983832	BovineHD2300009627	BovineHD2300009628	9.38	2.76 × 10^−7^	GPLD1	WITHIN
**HCT ****	2	23	32982199	32983832	BovineHD2300009627	BovineHD2300009628	10.37	6.55 × 10^−8^	GPLD1	WITHIN
**PLT ***	2	24	7751939	7753085	BovineHD2400002173	ARS-BFGL-NGS-112539	7.69	5.51 × 10^−6^	LOC100336967	WITHIN
**OTHR ***	5	8	17716845	17825120	BovineHD4100006566	ARS-BFGL-NGS-10089	8.43	9.50 × 10^−6^	LOC785941	8868
**RDWSD ***	7	23	31504452	31558355	BovineHD2300009102	BovineHD2300009103	6.43	8.17 × 10^−6^	HIST1H1E,HIST1H2BI	WITHIN
**PLCR ***	2	23	32935769	32941851	BovineHD2300009596	BovineHD2300009599	19.08	2.53 × 10^−6^	LOC520603	WITHIN

^1^ The abbreviations of hematological traits are given in Table 1, e.g., RBC is red blood cell count. ^2^ Number of SNPs included in haplotype block. ^3,4,5^ Chromosomal locations and positions of the most significant SNP associated with hematological traits in Bos taurus UMD_3.1 assembly, and starting position and ending position on the chromosome. ^6,7^ The first SNP and the second SNP in the haplotype block. ^8^ Interpretation rate of phenotypic variation of the marker. ^9^ The nearest annotated gene to the significant SNP. The annotated gene database is from http://asia.ensembl.org/index.html. SNP designated as in a gene or distance (bp) from a gene region in Bos taurus UMD_3.1 assembly. ** Genome-wide significant. * Chromosome-wide significant.

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
