# Peer review of "Analysis of Hematological Traits in Polled Yak by Genome-Wide Association Studies Using Individual SNPs and Haplotypes"

_genes, 2019, doi:10.3390/genes10060463_

Round 1
Reviewer 1 Report
PDF file lacks the tables and supplementary data needed to review the paper.
Author Response
Dear Editor and Reviewers,
Thank you very much for handling our manuscript entitle “Analysis of Hematological Traits in Polled Yak by Genome-wide Association Studies Using Individual SNPs and Haplotypes”.
We appreciate the comments from you and the two expert reviewers. We have evaluated the comments carefully. We have tried our best to address these concerns and comments. As a result, we believe that it will lead to a significant improvement of the manuscript. And the point-by-point response to the comments (in red) is included in this cover letter. In the revised manuscript, we have marked all the changed sentences or words as red in the text.
The following is a point-to-point response to the two reviewers’ comments.
Reviewer:
We wish to thank reviewer 1 for his/her support and insightful comments.
Major revisions 1:
PDF file lacks the tables and supplementary data needed to review the paper.
Response
Thank you for pointing out the problem of the article. I have attached the figure and tables in cover letter.
Table 1:
Descriptive statistics of 15 hematological traits in the YAK population.
Traits | Abbreviation | Value1 (No. 2) | CV3 |
Mean corpuscular hemoglobin concentration (g/L) | MCHC | 318.061±9.281 (313) | 2.92 |
Red blood cell volume distribution width-SD (fL) | RDWSD | 31.427±2.362 (315) | 7.52 |
Mean corpuscular volume (fl) | MCV | 42.725±3.25 (315) | 7.61 |
Red cell distribution coefficient of variation | RDWCV | 0.187±0.015 (315) | 7.75 |
Mean platelet volume (fl) | MPV | 7.146±0.627 (171) | 8.77 |
Mean corpuscular hemoglobin (pg) | MCH | 13.611±1.312 (315) | 9.64 |
Red blood cell count (1012/L) | RBC | 10.031±1.266 (314) | 12.62 |
Hematocrit (%) | HCT | 0.429±0.056 (309) | 12.83 |
Hemoglobin (g/L) | HGB | 136.217±18.577 (314) | 13.64 |
Platelet distribution width (%) | PDW | 8.777±1.314 (169) | 14.97 |
White blood cell count (109/L) | WBC | 9.04±2.197 (278) | 24.31 |
Medium white blood cell count (109/L) | OTHR | 4.603±1.505 (270) | 32.69 |
Lymphocyte count (109/L) | LYM | 4.538±1.588 (276) | 34.99 |
Platelet count (109/L) | PLT | 326.193±170.149 (311) | 52.17 |
Platelet-large cell ratio (%) | PLCR | 0.068±0.039 (111) | 57.71 |
1 Values are shown in mean ± standard deviation.
2 The numbers of recorded individuals are given in parentheses.
3 C.V: coefficient of variation.
Table 2:
Associated SNPs and nearby candidate genes for hematological traits.
Traits1 | Snp2 | CHr3 | Position4 | P value | Var (%)5 | Nearest gene6 | |
Name | Distance (bp) | ||||||
HCT* | BovineHD0500023978 | 5 | 84784989 | 8.51e-6 | 7.44 | IFLTD1 | 69046 |
HCT** | BovineHD4100016160 | 23 | 32994655 | 6.21e-8 | 10.48 | GPLD1 | within |
HCT** | BovineHD2300009636 | 23 | 32996521 | 6.55e-8 | 10.38 | GPLD1 | within |
HGB** | BovineHD4100016160 | 23 | 32994655 | 2.63e-7 | 9.48 | GPLD1 | within |
HGB** | BovineHD2300009636 | 23 | 32996521 | 2.76e-7 | 9.38 | GPLD1 | within |
RBC* | BovineHD1400016753 | 14 | 60395118 | 9.25e-7 | 7.52 | LOC782927 | within |
RBC* | BovineHD2300009636 | 23 | 32996521 | 5.40e-6 | 7.59 | GPLD1 | within |
RBC* | BovineHD4100016160 | 23 | 32994655 | 5.84e-6 | 7.59 | GPLD1 | within |
RBC* | BovineHD2300001545 | 23 | 6375280 | 6.80e-6 | 7.45 | C23H6ORF142 | 42 |
RBC* | BovineHD2300001550 | 23 | 6383172 | 7.65e-6 | 7.38 | C23H6ORF142 | 42 |
MPV* | BovineHD1000028551 | 10 | 98690434 | 3.00e-6 | 14.31 | LOC788481 | 13887 |
MPV* | BovineHD2000007009 | 20 | 23305743 | 5.68e-6 | 11.73 | IL31RA | 63 |
PDW* | BovineHD1700003099 | 17 | 10778156 | 1.98e-6 | 13.35 | EDNRA | within |
PLT* | BovineHD1100022362 | 11 | 77897626 | 1.68e-6 | 7.29 | APOB | within |
PLT* | BovineHD2100007261 | 21 | 25044956 | 3.87e-7 | 9.3 | SH3GL3 | 11156 |
PLT* | BovineHD2400002733 | 24 | 9663513 | 5.51e-6 | 7.69 | LOC538958 | 77309 |
LYM* | BovineHD2000004120 | 20 | 12840434 | 2.75e-6 | 9.13 | LOC529061 | within |
LYM* | BovineHD2100017671 | 21 | 61024785 | 4.74e-6 | 8.76 | LOC781495 | within |
LYM* | BovineHD2300005782 | 23 | 21870423 | 1.11e-6 | 8.48 | MUT | 68056 |
OTHR* | BovineHD0100040184 | 1 | 140543231 | 7.71e-7 | 10.17 | ATP2C1 | 20601 |
OTHR* | BovineHD0300001040 | 3 | 3461567 | 6.51e-6 | 8.7 | LOC100295170 | 2804 |
OTHR* | BovineHD0300021395 | 3 | 73543262 | 4.99e-6 | 8.89 | NEGR1 | within |
OTHR* | BovineHD0800005780 | 8 | 18417107 | 9.50e-6 | 8.44 | TUSC1 | 151827 |
1 The abbreviations of hematological traits are given in Table 1. e.g. MCV is Mean corpuscular volume.
2 The number of significant SNPs for each hematological trait.
3,4 Chromosomal locations and positions of the most significant SNP associated with hematological traits in Bos taurus UMD_3.1 assembly.
5 Interpretation rate of phenotypic variation of the marker
6 The nearest annotated gene to the significant SNP. The annotated gene database is from http://asia.ensembl.org/index.html. SNP designated as in a gene or distance (bp) from a gene region in Bos taurus UMD_3.1 assembly.
** Genome-wide significant.
* Chromosome-wide significant.
Table 3:
Associated haplotype blocks and nearby candidate genes for hematological traits.
Traits1 | Nsnp2 | CHr3 | Start4 | END5 | SNP16 | SNP27 | P value | Nearest gene8 | |
Name | Distance (bp) | ||||||||
RBC* | 2 | 23 | 6045141 | 6051410 | BovineHD2300001442 | BovineHD2300001444 | 7.16E-06 | TINAG | WITHIN |
RBC* | 2 | 23 | 6045141 | 6051410 | BovineHD2300001442 | BovineHD2300001444 | 7.45E-06 | TINAG | within |
RBC* | 2 | 23 | 32982199 | 32983832 | BovineHD2300009627 | BovineHD2300009628 | 5.40E-06 | GPLD1 | within |
HGB** | 2 | 23 | 32982199 | 32983832 | BovineHD2300009627 | BovineHD2300009628 | 2.76E-07 | GPLD1 | within |
HCT** | 2 | 23 | 32982199 | 32983832 | BovineHD2300009627 | BovineHD2300009628 | 6.55E-08 | GPLD1 | within |
PLT* | 2 | 24 | 7751939 | 7753085 | BovineHD2400002173 | ARS-BFGL-NGS-112539 | 5.51E-06 | LOC100336967 | within |
OTHR1* | 5 | 8 | 17716845 | 17825120 | BovineHD4100006566 | ARS-BFGL-NGS-10089 | 9.50E-06 | LOC785941 | 8868 |
RDWSD* | 7 | 23 | 31504452 | 31558355 | BovineHD2300009102 | BovineHD2300009103 | 8.17E-06 | HIST1H1E,HIST1H2BI | within |
PLCR* | 2 | 23 | 32935769 | 32941851 | BovineHD2300009596 | BovineHD2300009599 | 2.53E-06 | LOC520603 | within |
1 The abbreviations of hematological traits are given in Table 1. e.g.RBC is red blood cell count.
2 Number of SNPs included in haplotype block
3,4,5 Chromosomal locations and positions of the most significant SNP associated with hematological traits in Bos taurus UMD_3.1 assembly.and Starting position and ending position on the chrompsomal.
6,7 The first SNP and the second SNP in the haplotype block
8 The nearest annotated gene to the significant SNP. The annotated gene database is from http://asia.ensembl.org/index.html. SNP designated as in a gene or distance (bp) from a gene region in Bos taurus UMD_3.1 assembly.
** Genome-wide significant.
* Chromosome-wide significant.
* Chromosome-wide significant.
Table s1 Description of correlation and P-valueamong the hematological traits.
WBC | RBC | HGB | HCT | MCV | MCH | MCHC | PLT | LYM | OTHR | RDWSD | RDWCV | PDW | MPV | PLCR | ||
WBC | 1 | 0.526462551 | 0.58644072 | 0.515094335 | 0.611651747 | 0.61620158 | 0.556976699 | 0.315242373 | 0.880064287 | 0.857290084 | 0.6027535 | 0.505964913 | 0.124193831 | 0.127346684 | 0.169946988 | |
RBC | 0.526462551 | 1 | 0.960487027 | 0.925848517 | 0.875578577 | 0.849661918 | 0.87109631 | 0.447097153 | 0.4591237 | 0.469513992 | 0.880167171 | 0.91073657 | 0.197554853 | 0.182825912 | 0.048548976 | |
HGB | 0.58644072 | 0.960487027 | 1 | 0.943154457 | 0.928612594 | 0.921632078 | 0.873015693 | 0.413360269 | 0.511920396 | 0.529248144 | 0.918265069 | 0.846738778 | 0.223885793 | 0.215772123 | 0.194582093 | |
HCT | 0.515094335 | 0.925848517 | 0.943154457 | 1 | 0.864371281 | 0.850971417 | 0.821676123 | 0.390235064 | 0.45097453 | 0.46215879 | 0.855782693 | 0.810812499 | 0.189042401 | 0.183718153 | 0.135431852 | |
MCV | 0.611651747 | 0.875578577 | 0.928612594 | 0.864371281 | 1 | 0.995789804 | 0.946330707 | 0.471071163 | 0.542117269 | 0.542431436 | 0.986469314 | 0.896313463 | 0.320775837 | 0.334048876 | 0.305588765 | |
MCH | 0.61620158 | 0.849661918 | 0.921632078 | 0.850971417 | 0.995789804 | 1 | 0.934092986 | 0.454131684 | 0.546140464 | 0.547247794 | 0.980040989 | 0.871433012 | 0.330123796 | 0.344406505 | 0.335377317 | |
MCHC | 0.556976699 | 0.87109631 | 0.873015693 | 0.821676123 | 0.946330707 | 0.934092986 | 1 | 0.476511401 | 0.48695895 | 0.494376925 | 0.946764061 | 0.931171643 | 0.336324953 | 0.345931501 | 0.223890976 | |
PLT | 0.315242373 | 0.447097153 | 0.413360269 | 0.390235064 | 0.471071163 | 0.454131684 | 0.476511401 | 1 | 0.278859419 | 0.302872942 | 0.48108233 | 0.505262547 | 0.459604147 | 0.487394489 | 0.263216396 | |
LYM1 | 0.880064287 | 0.4591237 | 0.511920396 | 0.45097453 | 0.542117269 | 0.546140464 | 0.48695895 | 0.278859419 | 1 | 0.56330672 | 0.532398258 | 0.448735796 | 0.110523129 | 0.109593025 | 0.133118577 | |
OTHR1 | 0.857290084 | 0.469513992 | 0.529248144 | 0.46215879 | 0.542431436 | 0.547247794 | 0.494376925 | 0.302872942 | 0.56330672 | 1 | 0.537022686 | 0.438934021 | 0.136976194 | 0.144942624 | 0.185697461 | |
RDWSD | 0.6027535 | 0.880167171 | 0.918265069 | 0.855782693 | 0.986469314 | 0.980040989 | 0.946764061 | 0.48108233 | 0.532398258 | 0.537022686 | 1 | 0.926058888 | 0.300161286 | 0.311858646 | 0.269368136 | |
RDWCV | 0.505964913 | 0.91073657 | 0.846738778 | 0.810812499 | 0.896313463 | 0.871433012 | 0.931171643 | 0.505262547 | 0.448735796 | 0.438934021 | 0.926058888 | 1 | 0.270970566 | 0.273602533 | 0.069800483 | |
PDW | 0.124193831 | 0.197554853 | 0.223885793 | 0.189042401 | 0.320775837 | 0.330123796 | 0.336324953 | 0.459604147 | 0.110523129 | 0.136976194 | 0.300161286 | 0.270970566 | 1 | 0.974121831 | 0.604277102 | |
MPV | 0.127346684 | 0.182825912 | 0.215772123 | 0.183718153 | 0.334048876 | 0.344406505 | 0.345931501 | 0.487394489 | 0.109593025 | 0.144942624 | 0.311858646 | 0.273602533 | 0.974121831 | 1 | 0.651282263 | |
PLCR | 0.169946988 | 0.048548976 | 0.194582093 | 0.135431852 | 0.305588765 | 0.335377317 | 0.223890976 | 0.263216396 | 0.133118577 | 0.185697461 | 0.269368136 | 0.069800483 | 0.604277102 | 0.651282263 | 1 |
Figure 1 A portion of GWAS results of hematological traits at a significant level in Polled yak. X axis indicates chromosome, Y axis indicates -lg(P-value), the solid line indicates genome-wide significant level and the dashed line indicates chromosome-wide significant level, the bars below each chromosome are the density of the markers. RBC: red blood cell count. HCT: hematocrit; HGB: hemoglobin; OTHR: medium white blood cell count; PLT: platelet count; MPV: mean platelet volume; PLCR: platelet count; LYM: lymphocyte count; PDW: platelet distribution width.
Figure 2 Manhattan plots for the haplotype analysis of hematological traits surpass genome-wide significant threshold. log 10 (1/P-value) values are shown for all SNPs that passed quality control. The numbers indicate the chromosomes in the genome. The solid line and dotted line denotes the Bonferroni-corrected genome-wide and suggestive significant threshold, respectively. SNPs surpassing the genome-wide threshold are highlighted in red and SNPs reaching the suggestive threshold in green. the bars below each chromosome are the density of the markers. Eg Manhattan of HCT means Manhattan map obtained by haplotype analysis of HCT traits. HCT: hematocrit; RBC: red blood cell count; HGB: hemoglobin; OTHR: medium white blood cell count; PLT: platelet count; RDW-SD: red blood cell volume distribution width-SD; PLCR: platelet count.
We have tried our best to improve the manuscript according to the comments of the reviewers. We hope that we have addressed the reviewers’ comments to a satisfactory level and looking forward to the acceptance and publication of our manuscript in Genes. Finally, we would like to mention again that we deeply appreciate the editor’s and reviewers’ help.
Sincerely,
Ping Yan, Ph.D.
Professor, Animal Genetics, Breeding and Reproduction
Lanzhou Institute of Husbandry and Pharmaceutical Sciences
Chinese Academy of Agricultural Sciences
NO.335 Jiangouyan, Qilihe District Lanzhou 730050, Gansu, P. R. C
Tel: 0931-2115288
Email:pingyanlz@163.com

Reviewer 2 Report
Dear authors,
This is a GWAS study providing genetic markers potentially regulating hematological traits in a domestic animal living in high-altitude plateaus; Yak (Bos grunniens). I have a few comments to improve the manuscript for publication.
1- For single marker GWAS (line 118), what variables did you control for in the linear model?
2- Why did the authors prefer to use Tassel over PLINK for single marker analysis?
3- I would like to know a bit about your data, were they normally distributed?
4- The authors used loose parameters such as minor allele frequency (MAF) > 0.01 and call rate > 95%. Can you check, at least, the significant markers whether they are common variants (MAF > 0.05) and have high call rates (> 97%)?
5- Line 164, what do you mean by “OTHER”? Please make it clear for the reader.
6- Lines 184 through 189, the authors mentioned that single marker analysis outperforms the haplotype, whereas other studies showed the opposite. Please refer to studies that agree with your findings and also those which show the opposite.
7- Line 193, “when single marker analysis was used, and on chromosome 7 when haplotype analysis was employed” authors did not mention the number of SNPs on chromosome 7.
8- Line 208 & 209, HGB and HCT were mentioned in full in lines 95 and 96. So, there is no need to rewrite the full name.
Author Response
Dear Editor and Reviewers,
Thank you very much for handling our manuscript entitle “Analysis of Hematological Traits in Polled Yak by Genome-wide Association Studies Using Individual SNPs and Haplotypes”.
We appreciate the comments from you and the two expert reviewers. We have evaluated the comments carefully. We have tried our best to address these concerns and comments. As a result, we believe that it will lead to a significant improvement of the manuscript. And the point-by-point response to the comments (in red) is included in this cover letter. In the revised manuscript, we have marked all the changed sentences or words as red in the text.
The following is a point-to-point response to the two reviewers’ comments.
Reviewer:
We wish to thank reviewer 2 for his/her support and insightful comments.
Major revisions 1:
For single marker GWAS (line 118), what variables did you control for in the linear model?
Response:
Thank you for your comments.
y = Xbi+ Qv + e
y is the phenotype
X is the vector of the corresponding SNP genotype indicators
bi is the regression coefficient of phenotype on SNP genotypes
v is the effect of population structure
Q is the corresponding principal components matrix
e is the vector of residual errors with e, N (0, Iσ e2 )
The revised content is in lines 122 to 127 of the text.(page 3)
Major revisions 2:
Why did the authors prefer to use Tassel over PLINK for single marker analysis?
Response:
Thank you for your comments. Compared with plink, tassel is a graphical interface and easy to operate. It introduces phenotypes, and the genotype and Q matrix are simple and intuitive. And the software usage of the two is similar, and the correlation analysis between single mark and haplotype can be calculated. Therefore, tassel is selected as the data processing tool in this experiment.
Major revisions 3:
I would like to know a bit about your data, were they normally distributed?
Response:
Raw data link:https://figshare.com/s/efb004ad2335f96b3301
Major revisions 4:
The authors used loose parameters such as minor allele frequency (MAF) > 0.01 and call rate > 95%. Can you check, at least, the significant markers whether they are common variants (MAF > 0.05) and have high call rates (> 97%)?
Response:
Thank you for your suggestion, but the results obtained through software re-control according to your suggested conditions show that the available mark is less than the previous condition.which has a certain impact on the subsequent analysis.
I also refer to the article using the same method as this article, usually the MAF value is less than 0.01, and the call rate > 90%.
1. Zhang, F.; Zhang, Z.; Yan, X.; Chen, H.; Zhang, W.; Yuan, H.; Huang, L. Genome-wide association studies for hematological traits in Chinese Sutai pigs. Bmc Genetics 2014, 15, 41-41.
2. Wu, Y.; Fan, H.; Wang, Y.; Zhang, L.; Gao, X.; Chen, Y.; Li, J.; Ren, H.; Gao, H. Genome-wide association studies using haplotypes and individual SNPs in Simmental cattle. PloS one 2014, 9, e109330.
3. Thompson-Crispi, K.A.; Sargolzaei, M.; Ventura, R.; Abo-Ismail, M.; Miglior, F.; Schenkel, F.; Mallard, B.A. A genome-wide association study of immune response traits in Canadian Holstein cattle. Bmc Genomics 2014, 15, 559.
4. Stratz, P.; Schmid, M.; Wellmann, R.; Preuß, S.; Blaj, I.; Tetens, J.; Thaller, G.; Bennewitz, J. Linkage disequilibrium pattern and genome‐wide association mapping for meat traits in multiple porcine F2 crosses. Animal genetics 2018, 49, 403-412.
5. Mastrangelo, S.; Sottile, G.; Sutera, A.; Di Gerlando, R.; Tolone, M.; Moscarelli, A.; Sardina, M.; Portolano, B. Genome‐wide association study reveals the locus responsible for microtia in Valle del Belice sheep breed. Animal genetics 2018, 49, 636-640.
6. Jiang, S.; Xu, H.; Shen, Z.; Zhao, C.; Wu, C. Genome‐wide association analysis reveals novel loci for hypoxia adaptability in Tibetan chicken. Animal genetics 2018, 49, 337-339.
Major revisions 5 :
Line 164, what do you mean by “OTHER”? Please make it clear for the reader.
Response:
Thank you for reminding us to recheck Line 164,Among them, “OTHR” has written the full name in Line 100 . We found that “OTHER” is a writing mistake, we have modified the text to be more clear.
Major revisions 6
Lines 184 through 189, the authors mentioned that single marker analysis outperforms the haplotype, whereas other studies showed the opposite. Please refer to studies that agree with your findings and also those which show the opposite.
Response:
Thank you for pointing this out. The previous statement in the text is not clear, which leads to misunderstandings in the reading of the article. We have re-write this part to the reviewer's suggestion(line 189-192 page 5). According to your opinion, after reviewing the relevant literature, it is found that in the research and analysis of complex diseases, whether based on a single SNP locus or multiple SNP loci, in many cases, it is not as good as haplotype-based research [1] [2] [3] [4] [5] . The main advantages of using haplotypes for complex traits are three: 1)The haplotype is sometimes the basic unit of biological function; 2) the haplotype is a characteristic structure on the chromosome, which may exist in the form of a block and can be transmitted to the next generation as a whole; 3) statistically, using a haplotype It can reduce the dimensionality of research problems . In summary, the haplotype-based analysis method will have a long-term development in the field of finding and locating genes for complex diseases, and will continue to play an important role.
One disadvantage of haplotype analysis is the reduction in detection efficiency, which is greater than the single-point analysis. Zhang et al [6] also pointed out that this phenomenon is due to the degree of freedom. However, it is often difficult to weigh between increasing LD and increasing freedom resulting in reduced detection efficiency. Furthermore, the LDs distributed in the genome are not uniform, in some places the LD is high, and in some places the LD is low, so we compare the two methods to capture more associated SNPs.
1. Schaid, D.J. Evaluating associations of haplotypes with traits. Genetic Epidemiology: The Official Publication of the International Genetic Epidemiology Society 2004, 27, 348-364.
2. Schaid, D.J.; Rowland, C.M.; Tines, D.E.; Jacobson, R.M.; Poland, G.A. Score tests for association between traits and haplotypes when linkage phase is ambiguous. The American Journal of Human Genetics 2002, 70, 425-434.
3. Morris, R.W.; Kaplan, N.L. On the advantage of haplotype analysis in the presence of multiple disease susceptibility alleles. Genetic Epidemiology: The Official Publication of the International Genetic Epidemiology Society 2002, 23, 221-233.
4. Akey, J.; Jin, L.; Xiong, M. Haplotypes vs single marker linkage disequilibrium tests: what do we gain? European Journal of Human Genetics 2001, 9, 291.
5. Clark, A.G. The role of haplotypes in candidate gene studies. Genetic Epidemiology: The Official Publication of the International Genetic Epidemiology Society 2004, 27, 321-333.
6. Zhang, Z.; Guillaume, F.; Sartelet, A.; Charlier, C.; Georges, M.; Farnir, F.; Druet, T. Ancestral haplotype-based association mapping with generalized linear mixed models accounting for stratification. Bioinformatics 2012, 28, 2467-2473.
Major revisions 7:
Line 193, “when single marker analysis was used, and on chromosome 7 when haplotype analysis was employed” authors did not mention the number of SNPs on chromosome 7.
Response:
We appreciate the reviewer`s attention to the detail, and we have corrected the text as your suggested. “located at 24777963 bp on chromosome 7” is modified in the Line195-196 of the text. (page 5)
Major revisions 8:
Line 208 & 209, HGB and HCT were mentioned in full in lines 95 and 96. So, there is no need to rewrite the full name.
Response:
Thank you for your suggestion, I have removed the duplicate description in the text. (page 5, line 210)
We have tried our best to improve the manuscript according to the comments of the reviewers. We hope that we have addressed the reviewers’ comments to a satisfactory level and looking forward to the acceptance and publication of our manuscript in Genes. Finally, we would like to mention again that we deeply appreciate the editor’s and reviewers’ help.
Sincerely,
Ping Yan, Ph.D.
Professor, Animal Genetics, Breeding and Reproduction
Lanzhou Institute of Husbandry and Pharmaceutical Sciences
Chinese Academy of Agricultural Sciences
NO.335 Jiangouyan, Qilihe District Lanzhou 730050, Gansu, P. R. C
Tel: 0931-2115288
Email:pingyanlz@163.com

Round 2
Reviewer 1 Report
The genome-wide association studies (GWAS) utilizing a single marker or haplotype method to find markers associated with 15 hematological traits in yaks. They detect genes associated with 6 traits in 5 genes. This study aims at finding genetic markers for hematological which can be used as markers for blood characteristics and diseases in yaks. There are few concerned related to the study:
1. What is the significance of chromosome-wide significant SNP?
2. Authors should provide 51 single markers that were detected significant which were located within 35 annotated genes.
3. Figure 2. Resolution is bad so assessment is difficult.
4. Line 258/266: Difficult for the reviewer to understand.
5. Authors should calculate effect sizes and direction of effect of the markers.
6. The multiple testing was employed on all markers without taking into account LD structure, which might the number of independent tests performed.
7. The authors do not discuss the effect of the trait associated markers on the proteins. They do discuss the functional effects based on the findings from other species. But understanding the effect of the markers on the yak proteins is important. The reporting of gene function is not adding any knowledge to the field.
8. The authors may also discuss the relationships between different traits and how having markers for both in the same genome might affect the total health of the animal.
Author Response
Dear Editor and Reviewers,
Thank you very much for handling our manuscript entitled “Analysis of Hematological Traits in Polled Yak by Genome-wide Association Studies Using Individual SNPs and Haplotypes”.
We appreciate the comments from you and the two expert reviewers. We have evaluated the comments carefully. We have tried our best to address the concerns and comments. We believe that these comments will lead to a significant improvement in our manuscript. We have given point-to-point response to the comments (in red) in this cover letter. In the revised manuscript, we have marked all the changed sentences or words with red colour.
The following are point-to-point response to the two reviewers’ comments.
Reviewer:
We wish to thank reviewer 1 for his/her support and insightful comments.
Major revisions 1:
What is the significance of -wide significant SNP?
Response:
Thank you for your comments. The P-value threshold for genome-wide significance was determined by the Bonferroni method, in which P-value was divided by the number of tests [1].A SNP was considered to have the stringent genome-wide significance at P < 0.05/N, and have the suggestive (chromosome) significance at P < 1/N, where N stands for the number of SNPs in the data analysis. This concept is usually used in other articles [2-7]. The number of markers for each staining were sort out and used the above method i.e., 0.05/N (number of markers per chromosome).
chr | nub | pvaule(chrom-wide sig) | -log10(pvalue) |
1 | 6902 | 7.24428E-06 | 5.140004951 |
2 | 4770 | 1.04822E-05 | 4.979548375 |
3 | 7307 | 6.84275E-06 | 5.164769103 |
4 | 4058 | 1.23213E-05 | 4.909342038 |
5 | 6367 | 7.85299E-06 | 5.104964846 |
6 | 5479 | 9.12575E-06 | 5.039731296 |
7 | 3181 | 1.57183E-05 | 4.803593665 |
8 | 2177 | 2.29674E-05 | 4.638888425 |
9 | 6194 | 8.07233E-06 | 5.093001197 |
10 | 2732 | 1.83016E-05 | 4.737510691 |
11 | 2832 | 1.76554E-05 | 4.753123245 |
12 | 3760 | 1.32979E-05 | 4.876217841 |
13 | 2766 | 1.80766E-05 | 4.742882171 |
14 | 1954 | 2.55885E-05 | 4.591954555 |
15 | 5389 | 9.27816E-06 | 5.032538179 |
16 | 3423 | 1.46071E-05 | 4.835436895 |
17 | 5292 | 9.44822E-06 | 5.024649831 |
18 | 2508 | 1.99362E-05 | 4.700357528 |
19 | 1797 | 2.78242E-05 | 4.555578073 |
20 | 3285 | 1.52207E-05 | 4.81756537 |
21 | 2157 | 2.31803E-05 | 4.634880141 |
22 | 2684 | 1.86289E-05 | 4.729812507 |
23 | 5939 | 8.41893E-06 | 5.074743321 |
24 | 2333 | 2.14316E-05 | 4.668944734 |
25 | 2191 | 2.28206E-05 | 4.641672373 |
26 | 1406 | 3.55619E-05 | 4.449015316 |
27 | 1916 | 2.6096E-05 | 4.5834255 |
28 | 951 | 5.25762E-05 | 4.279210513 |
29 | 1731 | 2.8885E-05 | 4.539327064 |
1.Yang, Q., Cui, J., Chazaro, I., Cupples, L. A., & Demissie, S. (2005, December). Power and type I error rate of false discovery rate approaches in genome-wide association studies. In BMC genetics (Vol. 6, No. 1, p. S134). BioMed Central.
2. Ding, R., Yang, M., Wang, X., Quan, J., Zhuang, Z., Zhou, S., ... & Liu, D. (2018). Genetic architecture of feeding behavior and feed efficiency in a Duroc pig population. Frontiers in genetics, 9, 220.
3.Quan, J., Ding, R., Wang, X., Yang, M., Yang, Y., Zheng, E., ... & Yang, J. (2018). Genome-wide association study reveals genetic loci and candidate genes for average daily gain in Duroc pigs. Asian-Australasian journal of animal sciences, 31(4), 480.
4.Ding, R., Quan, J., Yang, M., Wang, X., Zheng, E., Yang, H., ... & Liu, D. (2017). Genome-wide association analysis reveals genetic loci and candidate genes for feeding behavior and eating efficiency in Duroc boars. PloS one, 12(8), e0183244.
5.Ji, J., Zhou, L., Guo, Y., Huang, L., & Ma, J. (2017). Genome-wide association study identifies 22 new loci for body dimension and body weight traits in a white Duroc× Erhualian F2 intercross population. Asian-Australasian journal of animal sciences, 30(8), 1066.
6.Ma, J., Yang, J., Zhou, L., Ren, J., Liu, X., Zhang, H., ... & Xing, Y. (2014). A splice mutation in the PHKG1 gene causes high glycogen content and low meat quality in pig skeletal muscle. PLoS genetics, 10(10), e1004710.
7.Ma, J., Yang, J., Zhou, L., Zhang, Z., Ma, H., Xie, X., ... & Liu, X. (2013). Genome-wide association study of meat quality traits in a White Duroc× Erhualian F2 intercross and Chinese Sutai pigs. PloS one, 8(5), e64047.
Major revisions 2:
Authors should provide 51 single markers that were detected significant which were located within 35 annotated genes.
Response:
Thank you for reminding us to recheck Line 157-158, we have corrected the text as you suggested. “Together, 16 single markers were detected which were located within 5 annotated genes.” And we have added all the prominent marker to the manuscript in the form of additional file (Table s2).
Major revisions 3:
Figure 2. Resolution is bad so assessment is difficult.
Response:
We have replaced the previous old version with a high resolution version as per your suggestion.
Major revisions 4:
Line 258/266: Difficult for the reviewer to understand.
Response:
We have revised the said paragraph and attached a relationship heat map for several types of traits. “The results of this GWAS showed a significant association pattern of erythrocyte phenotypes. The Manhattan plot pattern was similar for HGB, HCT, and RBC and the three erythrocyte traits which shared common regions at BTA23, from 32,982Mb to 32,983Mb containing two SNPs(BovineHD2300009627 and BovineHD2300009628). Both of these parameters are directly related to Erythrocytes, respectively correlation analysis of hematological traits by comparison (Additional file 1: Table S1). Moreover, the results indicated a high correlation between the three erythrocyte traits (r1=0.96, r2=0.93, r3=0.94; P<0.001). A significant degree of correlation implies that the QTL on BTA23 might be pleiotropic. However, this association was not found in leukocyte and platelet traits.” The revised part is on page 10 line 264-272 of the original text.
Major revisions 5:
Authors should calculate effect sizes and direction of effect of the markers.
Response:
In table 2, "Var (%)" in the header represents the effect size of the marker.
In the T1 screenshot, we have attached the results calculated by the tassel software and a description of each result. It was found that the use of the tassel GLM model did not result in the direction of the mark, so the data of the mark direction could not be provided in table 2 and table 3 in manuscript.
Screenshot“T1”
Major revisions 6:
The multiple testing was employed on all markers without taking into account LD structure, which might the number of independent tests performed.
Response:
We have used two methods in this experiment, the single-marker and haplotype methods, and both of these methods were used to capture more markers. The two methods do not have all coincident markers, on one hand because the sites you mentioned are not in the block, and on other hand, the density reduction after gene chip control is not associated.
Major revisions 7:
The authors do not discuss the effect of the trait associated markers on the proteins. They do discuss the functional effects based on the findings from other species. But understanding the effect of the markers on the yak proteins is important. The reporting of gene function is not adding any knowledge to the field.
Response:
We have corrected the text as you suggested. “There is a coordination relationship between the platelet activation mechanism and the cholesterol-rich microdomain. [1] In the species of cattle, close relatives of yak, the base mutation and insertion mutation of the APOB gene can be found to cause cholesterol deficiency in the individual [2] [3].”The revised part is on page 11 line 296-300 of the original text. Only the APOB gene related species and trait-related papers can be found in literature, while the other four genes are not available.
1. Bodin, S., Tronchère, H., & Payrastre, B. (2003). Lipid rafts are critical membrane domains in blood platelet activation processes. Biochimica et Biophysica Acta (BBA)-Biomembranes, 1610(2), 247-257.
2 .Menzi, F., Besuchet‐Schmutz, N., Fragnière, M., Hofstetter, S., Jagannathan, V., Mock, T., ... & Meylan, M. (2016). A transposable element insertion in APOB causes cholesterol deficiency in Holstein cattle. Animal genetics, 47(2), 253-257.
3. Gross, J. J., Schwinn, A. C., Schmitz-Hsu, F., Menzi, F., Drögemüller, C., Albrecht, C., & Bruckmaier, R. M. (2016). Rapid Communication: Cholesterol deficiency–associated APOB mutation impacts lipid metabolism in Holstein calves and breeding bulls. Journal of animal science, 94(4), 1761-1766.
Major revisions 8:
The authors may also discuss the relationships between different traits and how having markers for both in the same genome might affect the total health of the animal.
Response:
We have classified the blood traits used in the trials into three groups: White blood cell, Erythrocyte Traits, and platelets. And in the process of answering the "major revision 4" question, the relationship between the traits was analyzed by using a heat map for blood traits, and then described later.
We have tried our best to improve the manuscript according to the comments of the reviewers. We hope that we have addressed the reviewers’ comments to a satisfactory level and looking forward to the acceptance and publication of our manuscript in Genes. We deeply appreciate for your time and help.
Sincerely,
Ping Yan, Ph.D.
Professor, Animal Genetics, Breeding and Reproduction
Lanzhou Institute of Husbandry and Pharmaceutical Sciences
Chinese Academy of Agricultural Sciences
NO.335 Jiangouyan, Qilihe District Lanzhou 730050, Gansu, P. R. C
Tel: 0931-2115288
Email:pingyanlz@163.com
